# Formal Verification for Task Description Languages. A Petri Net Approach

**DOI:** 10.3390/s19224965

**Published:** 2019-11-14

**Authors:** Joaquín López, Alejandro Santana-Alonso, Miguel Díaz-Cacho Medina

**Affiliations:** Dep. Ingeniería de Sistemas y Automática, University of Vigo, 36200 Vigo, Spain; asantana@uvigo.es (A.S.-A.); mcacho@uvigo.es (M.D.-C.M.)

**Keywords:** autonomous robot system, task-level programming, formal verification, task description language

## Abstract

One of the main challenges in verifying robotic systems is its asynchronous interaction with an unstructured environment, observed by imperfect sensors. Autonomous robot systems usually require some language to support task-level control. This paper presents an effective approach to apply formal verification methods for that kind of language. A main contribution of this method is to avoid modeling the robotic system with a specific formalism. The approach translates the task-level control models into a Petri net (PN) based representation. This is used to define new methods to analyze some task properties such as liveness, deadlock-freeness and terminability. The approach has been applied to the Task Description Language (TDL) and it is illustrated by experiments. The final goal is to create new tools within the application development environment to include formal verification as part of the normal software development cycle. The TDL to PN translator uses the Petri Net Markup Language (PNML) as its file format. This format permits interoperability with other Petri net tools that can also be used to analyze the PNs.

## 1. Introduction

Autonomous robotic systems tend to be complex because they need to interact asynchronously and in real time with uncertain and dynamic environments. Task-level control programs in autonomous robot applications are usually complex, difficult to develop and debug [1]. Among other requirements, that kind of program has to deal with activities running concurrently, such as moving and sensing, planning and executing, etc. Besides, some actions need to be scheduled to execute at an absolute or relative time point. In other cases, the different execution phases of some activities might be restricted to the execution phases of other activities. For example, one action should be started after another action finishes. Another difficulty is that some actions might need to be monitored and some exceptions require to be handled. Also, some of these exceptions might need to be managed at different levels. For example, if the sensors of a robot detect an unexpected obstacle, it might first try to avoid it. If that fails, the robot could try again with a different obstacle avoidance method. If that still fails, it could try to find a new path or it might switch to another goal.

Implementing such executive functions using conventional programming languages would result in highly non-linear code that is often cumbersome and unclear, difficult to understand, debug and maintain. Instead, new languages have been developed, including: action packages (RAPs) [2], task description languages for AUVs [3], the Mission Control Language for AUVs [4], the execution support language (ESL) [5], the task description language (TDL) [1] and the plan execution interchange language (PLEXIL) [6]. All of these executive languages provide support for reliably achieving higher-level tasks and include several capabilities such as:
Support the simple serial and concurrent temporal constraints between tasks.Support for hierarchical decomposition of tasks into subtasks.Support for timeouts to trigger after waiting a specified amount of time.Support conditional and iterative execution of subtasks.Support for tools to handle exceptions and monitors.

However, due to the increased capability and flexibility of these languages, it is typically quite difficult to assess the reliability of the developed application [7]. It is very difficult to verify all possible sequences of events and the corresponding outcomes of the programs. Other methods to create reliable systems are model-checking techniques. These techniques use a formal language, such as PROMELA [8] or SMV [9], to define formal specifications that indicate the desirable properties of the system to be verified. The model checker then determines whether the properties hold under all the execution traces. The verification tool presented in [10], called ABV, model-checks the model against the requirements. The architecture is defined using the Architecture Analysis and Design Language (AADL) that contains a Behavior Annex for describing the behavior of an AADL model at an abstract level. The requirements are specified in Computation Tree Logic (CTL).

On the other hand, formal verification methods such as the one presented in this paper, verify whether the system holds a set of general properties but do not need the definition of specific requirements. Therefore, they are simpler to use as part of the normal software development cycle but they are not able to check the same kind of requirements.

Formal verification is a powerful tool that has been used successfully to assess the reliability of autonomy software. Besides, research on PNs has produced an extensive set of analysis tools that allows the study of liveness, boundedness, deadlock-freeness and other properties for control programs [11]. This research intends to use these results to analyze control programs, mainly robot control systems, written in some of the task programming languages mentioned before. The first step in this process is to model the system with Petri nets and the second step is to analyze the resulting Petri nets.

The TDL language was selected because it is one of the more complete task description languages that contains most of other languages features and supports a wide range of temporal constraints [1].

The main contribution of this research is to provide an automatic method to carry out the formal verification of the task-level control in autonomous robotic systems using Petri nets as the model-checking language. Unlike previous solutions, the model used for verification is obtained directly from the task-control program. The contributions include:
Formalization of a method that translates a task description language into PNs. To test the system, a program that translates TDL into Petri nets was implemented.The analysis of PNs has been extended to be applied in the verification of the hierarchical PNs obtained from the Task to PN translation step.New properties have been added to the classic PN analysis to formally verify the tasks of autonomous robot systems.Application of this proposal to the mobile robot project BellBot [12].

The long-term goal is to include formal verification as part of the normal software development cycle including it in the development environments tools (such as RIDE [13]). This new function will enable engineers to automatically analyze the task-level control programs.

The rest of the paper is organized as follows: the next section describes the related work. The basic concepts about PNs and TDL are defined in Section 3. The process to translate the main TDL statements into Petri nets is described in Section 4. The Petri net analysis methods proposed in this research are described in Section 5. Finally, Section 6 describes the results and Section 7 closes the paper with the conclusions.

## 2. Related Work

Formal specification and verification of robot control systems is a challenging topic because they are hybrid, complex systems that use a wide variety of sensors and actuators to interact with the physical world. They have to execute complex tasks while, at the same time, they need to detect and react to unexpected situations. Due to this complexity, testing and simulation alone are not sufficient to guarantee the reliability of robotic systems. To deal with this complexity, the control system is divided into interacting subsystems. This division establishes the robot control architecture. Even though there are many different control architectures [14], a common feature of all of them is the modular decomposition into simpler pieces. The research presented in this paper focuses on the executive part of the control architecture that deals with the higher-level tasks [14].

An extended survey on formal specification and verification of autonomous robotic systems is presented in [15]. Many formalisms have been used to specify or verify robotic systems. Some of the most popular tools specify properties in temporal logic such as UPPAAL [16] and PRISM [17]. Formalisms for discrete-event systems are also among the most widespread and include PNs [18], Time Automata (TA) [19], Finite-State Automata (FSA) [20] and Markov chains [21]. Process algebras such as Finite State Processes (FSP) [22] that define the system behavior in terms of processes interactions are another popular choice.

Since almost all the robotic systems use some architectural framework, some verification research has been focusing on a general approach based on the framework. In this case, the verification approach can be applied to any project that is developed using the same framework. For example, in [19] an approach to model and verify systems developed using the Robotic Operating System ROS [23] was proposed. The approach uses real time properties, focusing on one of the main features of ROS: the communication between nodes. The UPPAAL [24] model checker is used to model ROS applications and to verify real-time properties.

As it was mentioned in the introduction, many autonomous systems use special-purpose languages to define and execute reliable higher-level tasks. All of them provide support for hierarchical decomposition of tasks into subtasks and they provide conditional and iteration capabilities. Some of those languages are an extension of another language. For example, ESL is an extension of Lisp and TDL is an extension of C++. Other languages such as RAPs, PRS and PLEXIL do not extend other existing languages. All of them support sequential and parallel temporal constraints between tasks. The purpose of this research is to verify the robotic system using directly the programs written in these languages instead of having to specify again the system properties in a different formalism. Instead, a parser will make this conversion process automatically. This is, converting TDL into PNs.

PNs have also been used to model robotic plans and behaviors. In this case, the first step described in this research is omitted since the task programming language is the same as the verification formalism. For example, the research presented in [25] uses PNs to model single-robot tasks while the work presented in [26] uses the PNs to model multi-robot tasks. Both systems were evaluated through simulation while the approach for modeling multi-robot tasks presented in [27] was evaluated in real robots. In [13] the RoboGraph tool is used to define tasks as PNs and save them in xml files. Those files are loaded by a dispatcher when the execution of the corresponding task is required. The same tool is also used to monitor the execution of the task by showing the current state of the PN in a graphical interface.

The idea of using PNs as a formal language to verify different kinds of hardware and systems designs is not new. For example, formal verification in SystemC designs is presented in [28,29]. The solution in [29] includes an algorithm for automatic translation of MSC (Message Sequence Chart) diagrams compliant with MSC’2000 standard into Petri nets. However, this verification process using PNs has not been applied before to task programming languages.

A former approach for TDL formal verification using tools that automatically convert autonomy software into formal models is described in [7]. The formal models are verified using model-checking and the model-checking language was SMV. However, only a subset of TDL constructs can be handled omitting structures such as model metric temporal constraints and iterative constructs. Also, another major issue of that solution is that it is usually quite difficult to diagnose the error directly from the counterexamples produced by SMV.

## 3. Languages and Models Used in this Research

This section describes the basic concepts of TDL and PNs; a detailed description of them can be found in [1,30] respectively.

### 3.1. Task Description Language

TDL is an extension of C++ that provides syntactic support for task decomposition, synchronization, execution monitoring and exception handling [1]. A compiler transforms TDL into pure C++ code (Figure 1). The compiler utilizes a platform-independent task management library called TCM (Task Control Management). We want to provide this compiler the ability to analyze some properties of the possible execution traces of the control program. For that purpose, the TDL code is also transformed into Petri nets (PNML format [31]) and these Petri nets are analyzed to produce a formal verification report. This is shown in Figure 1 with the two dotted bottom boxes. We are mainly interested in detecting possible deadlocks and checking liveness and terminability. The PNML format can be used by several tools such as PIPE [32] or RoboGraph [33] to represent and edit Petri nets.

Before creating a translator from the TDL language into another formalism it is necessary to formalize the semantics of the language. TDL simplifies the process of specifying how concurrent robot tasks should behave and interact. TDL, which is based on the TCA (Task Control Architecture) [34], has been used to implement the executive layer of several autonomous mobile robot systems (e.g., [35]). While TDL is based on a general-purpose programming language (C++), here we only verify the parts of the language that are concerned with task decomposition and synchronization.

Listing 1 shows a simple mobile robot task written in TDL for the BellBot [12] application. The task includes delivering the newspaper to a list of rooms in a hotel. The TDL task (*deliverNewspaper*) starts executing an *init_all* subtask. After *init_all* finishes, the *goTo* subtask drives the robot to next room. Once the robot is in front of the room, executes concurrently the tasks *speak* and *hand_over*. The last sequence of three tasks (*goTo*, *speak* and *hand_over*) are repeated for every guest in the list.

Each **Goal** statement defined in Listing 1 is considered as a task that can be at the root of the task tree (*deliverNewspaper*) or as a child (*init_all*, *goTo*, *speak* and *hand_over*). A child node of the task tree can be added using the **spawn** statement (Listing 1). **spawn** is non-blocking, in that the child subtask may not actually be handled by the time control returns to the parent task.
Listing 1: TDL *deliverNewspaper* task definition (Simplified).
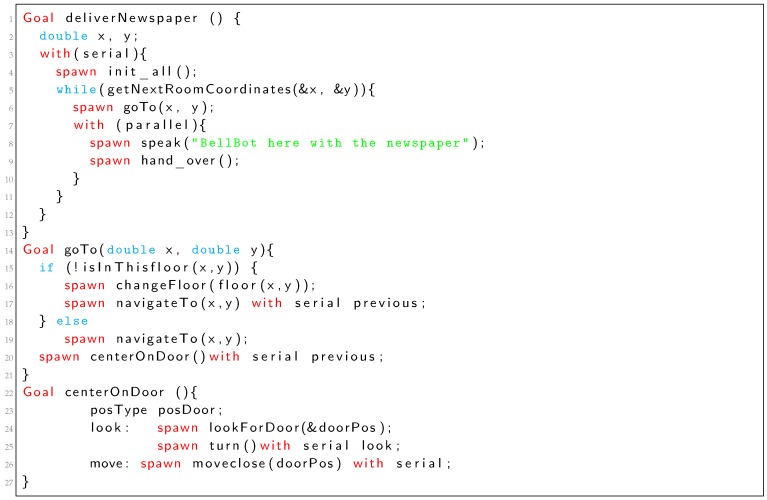


The **with** clause is used to synchronize spawned tasks, adding temporal restrictions with respect to other tasks. For example, tasks *init_all* and *goTo* in Listing 1 are executed in serial (*goTo* starts after *init_all* finishes). However, *speak* and *hand_over* are executed in parallel (at the same time). A detailed description of the syntax for the defined set of TDL constraints and statements can be found in [1].

### 3.2. Petri Nets

A Petri net is a model for the description of discrete event systems. For the sake of completeness, the basic notions of PNs are introduced in this section. A more detailed description can be found in [30]. Unlike other standards, PNs have an exact mathematical definition of their execution, with a well-developed mathematical theory for process analysis. A Petri net is a mathematical model with a graphical representation. Following the notation in [30], a Petri net is a 5-tuple PN = (*P*, *T*, *F*, *W*, $M_{0}$) where:
$P=\{p_{1},p_{2},p_{3},\ldots ,p_{n}\}$ is a finite set of places.$T=\{t_{1},t_{2},t_{3},\ldots ,t_{n}\}$ is a finite set of transitions.F⊆(P×T)∪(T×P) is a set of arcs (flow relation)W:F→{1,2,3,...} is a weight function that assigns a weight (positive nonzero integer) to each arc.$M_{0}:P\to \left\{1,2,3,...\right\}$ is the initial marking that assigns a positive number of tokens to each place.

A PN can be represented as a directed graph (Figure 2), where the nodes are the places (represented by circles) and transitions (represented by rectangles). The directed arcs join places with transitions. The tokens of a marking are represented by dots inside the circles (places). For example, the first place in the PN shown on the left part of Figure 2 has a token which is the only token in the initial marking.

The set of input places of a transition *t* (also known as pre-places of *t*) is defined as ${}^{*}t=\left\{p\in P|W\left(p,t\right)>0\right\}$. In the graphical representation, the input places of a transition are all the places from which starts an arc that ends in the transition. For example, in Figure 2, ${}^{*}t_{1}=\left\{p1\right\}$. In a similar way, the set of output places of a transition *t* (also known as post-places of *t*) is defined as $t^{*}=\left\{p\in P|W\left(t,p\right)>0\right\}$. For example, in Figure 2, $t_{1}^{*}=\left\{p2,p3\right\}$.

The Petri nets used here are binary which means that the tokens associated to the places can only be one or zero. Also, the weights (*W*) associated to each arc is always one and the net is therefore defined by the 4-tuple $N=\{P,T,F,M_{0}\}$.

PNs are used to model discrete event systems where the state of the system is represented by the net marking ($M_{i}$) that changes (dynamic behavior) according to the transition firing rules:A transition *t* is enabled if all its input places $p_{i}$ (i.e., $(p_{i},t)\in F$) have a token and all its post-places are empty. This is considered a safe enabling rule, while a general enabling rule does not verify the post-places. Since we only deal with binary PNs, a place should contain one token at most, no accumulation of tokens is allowed. Therefore the net design should grant the safe enabling rule.A transition fires if it is enabled and fulfills all its firing conditions. For Interpreted Petri nets (IPN (English IPN notation was taken from [30])). Ref. [36] the firing conditions can be associated to transitions. By default, an enabled transition without firing conditions is always evaluated to true. The condition associated to transition $t_{1}$ in Figure 2 is that both A and B should be true.The firing of a transition *t* removes a token from the input places and adds a token to all the output places. Figure 2 shows the changes on the marking of a PN after firing a transition.

The set of all reachable markings from $M_{0}$ is denoted by $R(M_{0})$. For Interpreted Petri nets (IPN) [34] an action can be associated to a place. In this case, whenever the place gets a token (mark) the action is executed.

## 4. Translating TDL into Petri Nets

Here, we are concerned only with the task decomposition and task synchronization aspects of TDL. A task in TDL is a section of C++ code that can include several robot actions and/or **spawn** subtasks. Constraints between TDL tasks are defined using a language of relational and metric temporal restrictions. The constraints can refer to various timing aspects of the task:The handling of a task is the time to execute the C++ code of the task.The expansion of a task is the handling of all non-leaf-node tasks. This is the time to expand the subtree rooted at that task.The execution of a task is the time to handle all leaf-node tasks.

TDL allows setting temporal constraints on the different aspects of the task. However, in this research we only consider the execution aspect of the task.

TDL-based control programs operate by creating and executing task trees that are generated dynamically. Every task tree node is a parameterized piece of code that carries out an action. An action can include classic computer computations, robot specific functions or dynamic operations in the task tree such as adding new nodes. Actions can include conditional, iterative and even recursive code. One example of iterative code is the **while** loop in line 3 of Listing 1 that repeats actions spawned within the loop (*goTo*, *speak* and *hand_over*). The **if** sentence in line 15 of Listing 1 is one example of conditional task execution.

Task trees are generated dynamically. Each task tree represents a single execution trace of the control program [1]. However, from the formal verification point of view, all the possible execution traces should be considered. Therefore, the Petri net model should include all the possible execution traces of the control program.

Instead of analyzing the entire control program at once, we use Hierarchical Petri nets to model different levels on the spawned tree. The main program that spawns different nodes (goals, commands, monitors and exceptions) is modeled as the Main (root) Petri net. Every goal, command, monitor and exception is also modeled in the corresponding Petri nets and analyzed independently.

This decomposition into simpler Petri nets preserves some of the properties that we want to verify on the task-level control programs (Section 5).

The process to obtain the corresponding Petri nets from the TDL code is done in two steps:In the first step, all statements including some **spawn** instruction are converted into the corresponding PN structure. Each statement (**spawn**, **if**, **while**, **for**, …) is translated into a PN structure and included in the goal structure.In the second step, the temporal constraints between **spawns** are stated in the PN. The algorithm used keeps a record of the previous spawns that can be used in future statements. For example, in case of an **if**, the previous structure might be a spawn on the **if** branch, the **else** branch or in the statement previous to the **if**.

In the following subsection the first step is presented. This is, the translation of the main TDL expressions into Petri nets.

### 4.1. Translating TDL Statements into Petri Net Structures

A parser programmed in Java using the Open Source parser generator JavaCC [37] was implemented. That parser translates TDL code into PNML code. The different TDL keywords are used by the parser to extract the main TDL expressions and to create an internal representation structure that is then translated into PNs. The transformation rules are embedded in the parser.

Every TDL goal is translated into a PN. TDL goals can spawn other goals using the **spawn** statement. This statement is translated into the PN as a *goal place* that represents the goal body and a *goal transition* that is going to be fired when the goal ends. For the execution of each goal, a new instance of the corresponding PN is created. When the *goal place* gets a token, the goal Petri net is started with the initial marking. The condition associated to the *goal transition* is that the goal Petri net reaches a final marking. The left thread (branch) of the Petri net in Figure 3 represents the “spawn init_all();” statement. The *goal place*
$p_{3}$ labeled as *init_all* represents the *init_all* goal body and transition t4 represents the *goal transition*.

Spawned tasks may be embedded within iterative or conditional code resulting in different task trees. Listing 1 shows one example of a task (*deliverNewspaper*) defined in TDL. It uses a **while** loop to repeat a deliver newspaper operation for all the guests (rooms) that requested the newspaper (getNextRoomCoordinates). Figure 3 shows the Petri net for the *deliverNewspaper* goal of Listing 1. Empty places (labelled as $p_{5}$ and $p_{10}$) represent the temporal constraints that will be explained in next subsection.

The **while** loop starts with the place labelled as $p_{6}$. If the **while** condition evaluates to true, transition $t_{6}$ is fired and actions inside the loop are executed. Inside the loop, the tasks *goTo*, *speak* and *hand_over* are executed. On the other case, if the condition evaluates to false, transition $t_{9}$ is fired and the loop is ended.

A **do-while** loop is translated in a similar way but in this case the evaluation of the condition takes place in a transition at the end of the loop.

A **for** loop is implemented in the same way because it follows the same pattern. Even though the initial and final conditions of the loop are different, the Petri net analysis is based on the structure of the Petri net but not the conditions associated to the transitions. For example, a blocked transition due to a wrong condition that never holds will not be detected by this kind of systems.

Statement **if** is represented with one “or node” (selection) in the PN. This node contains a place and two output transitions that start the **if** and **else** branches. Both branches should conclude in another “or node” (attribution) containing a place with two input transitions one for each branch. Each branch can include any set of statements. If there is no **else** statement, an empty place without actions will be used to represent the **else** case.

The task *goTo* of Listing 1 which Petri net is represented in Figure 4 includes an **if** statement. In the Petri net of Figure 4 the **if** structure starts with place $p_{1}$. Transition $t_{2}$ is fired when the **if** condition holds and goals *changeFloor* and *navigateTo* are spawned. Transition $t_{3}$ is fired when the **if** condition does not hold (case else) and in this case only *navigateTo* goal is spawned.

Listing 2 shows the implementation of the *getCloseToDoor* goal using a recursive spawn of itself. In this case, the robot executes recursive approaches to the door until it gets close enough (function *isCloseToDoor* returns true). Figure 5 shows the PN corresponding to Listing 2. The recursive execution takes place in p14 that represents a new execution of the *getCloseToDoor* goal.
Listing 2: TDL *getCloseToDoor* task definition (iterative example).
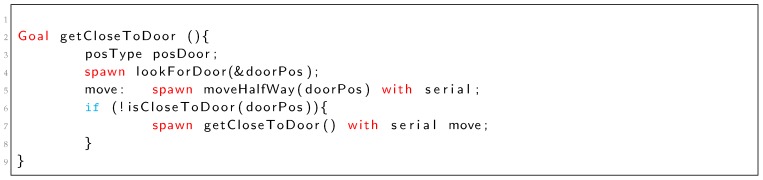


### 4.2. Adding Temporal Goal Constraints into Petri Nets

TDL provides a wide set of constraint statements to set temporal restrictions between tasks together with some shorthands for commonly used constraints. In the task *centerOnDoor* of Listing 1 the constraints are established in the spawn statements using the **with** statement. That is the most commonly used statement to represent task constraints. However, in order to set a constraint over several *spawns*, the **with** statement can also be used as in *deliverNewspaper* task.

By default, TDL tasks modeled as described in last section are executed in parallel and represented in parallel threads on the PN. Therefore, parallel constraints do not need to be added to the PNs. Serial constraints between two tasks are modeled by adding a new place to the Petri net. For example, in Figure 6 place $p_{3}$ represents a serial constraint between *turn* and *lookForDoor* goals. The constraint is established because transition $t_{3}$ is not enabled until place $p_{3}$ has a token, which means that *turn* goal has finished.

The **with** statement can be used with several tags that include some keywords (**self**, **parent**, or **previous**) or the the name of a spawned task that appear within the task body [1]. If multiple tasks of the same name are spawned, the referential ambiguity is resolved by using explicit labels. Label usage examples can be seen in task *centerOnDoor* of Listing 1 with labels “*look*” and “*move*”.

The tag **self** is used to reference the task that is being spawned and the tag **parent** is used to reference the enclosing task. The tag **previous** refers to the previous spawned task. Conditional or iterative code can include *spawn task* commands. In these cases it is impossible to determine the previous task statically, at compile time. For example, if the previous spawn is embedded in a conditional statement, the Petri net has to include the possibilities that the previous spawn might be in one branch of the condition or even in a previous spawn. For example, in Figure 4 we can see that goal *centerOnDoor* is restricted to the end of the two possible executions of the *navegateTo* goal.

In some situations, it is necessary to use complex PN structures in order to deal with all possible situations. For example, if we change the *goTo* goal according to the Listing 3, the corresponding PN is represented in Figure 7. Place $p_{5}$ represents the serial restriction between *locateRobot* and *changeFloor* (if condition evaluates to true) or between *locateRobot* and *navigateTo* (if condition evaluates to false). Places $p_{8}$, $p_{9}$, $p_{10}$, $p_{18}$, $p_{21}$ and transitions $t_{13}$ and $p_{14}$ represent the “*with serial previous*” restriction for the *navigateTo* goal (Listing 3).Listing 3: TDL Goal *goTo* new definition with different restrictions.
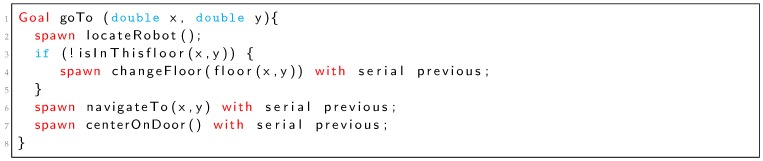


### 4.3. Limitations on the Petri Net Representations

Actions associated with nodes in the task tree can include conditional, iterative and even recursive code. Therefore, the same control program can result in widely different execution traces and different task trees. Petri nets that include all the possible traces can be quite complex and very difficult to analyze.

According to [38], Petri nets with inhibitor arcs allow arbitrary computations on numbers of tokens to be expressed, which makes the formalism Turing complete. However, for the sake of simplicity in representation and analysis, we have decided to use regular binary Petri nets since they cover most of the situations we have found in real applications. For that purpose, some restrictions in the statements that can be translated into Petri nets have been set:Loop iterations have to be executed in serial. This is, one iteration has to start after all the commands in the previous iteration finished. Without this restriction, the entire set of tasks spawned inside a loop can be executed in parallel. To represent this situation non-binary Petri nets should be used.Restrictions on spawn goals and commands inside a loop must refer only to goals and commands spawned inside a loop. In other words, restrictions between goals spawned inside the loop and goals spawned outside of loops are not translated into the Petri nets.

A study on the TDL software developed for different projects let us conclude that, even though TDL includes all these possibilities, very few programmers are actually using them. Therefore, even with these limitations, we are able to analyze most of the task-level control software developed in TDL.

## 5. Petri Net Analysis

Petri nets provide a balance between modeling power and analyzability because many properties about asynchronous systems can be automatically determined using Petri nets. The study of some Petri net properties has proved to be a powerful technique for investigating properties of the modeled systems. Two important criteria in the study of Petri net models are liveness and safeness.

Because of the nature of this verification problem, two issues need to be addressed before applying the traditional PN analysis. The first one comes from the fact that a PN can be started from another PN. This can be seen as hierarchical PNs [39] where the hierarchy is used to break down the complexity of the model. The second issue to be addressed is that most of the tasks and subtasks modeled here should finish. Therefore, the concept of terminability needs to be introduced.

### 5.1. Basic Definitions

One of the properties of the Petri nets that we will use later is liveness, which is related to the complete absence of deadlocks. Liveness indicates the capability of transitions to be fired again. A transition $t_{i}$ is said to be live with respect to an initial marking $M_{0}$ if, from any reachable marking $M_{j}\in R\left(M_{0}\right)$, there exists a sequence of transition firings that grants $t_{i}$ can be enabled and fired again. Otherwise, it is said to be dead. A PN is live if and only if all its transitions are said to be live. Therefore, a Petri net is said to be live if, no matter what marking has been reached from the initial marking, it is always possible to fire any transition of the net by progressing through some further firing sequence.

The concept of liveness is closely related to the absence of deadlocks. This means that a live Petri net guarantees deadlock-free operation, no matter what firing sequence is chosen. However, liveness is a more restrictive condition than deadlock-freeness.

As shown before, we use a hierarchical representation of Petri nets by obtaining a Petri net for the main program and all the Petri nets for the goals spawned in the main and subsequent goals. This decomposition into simpler Petri nets can simplify the analysis if we prove that these properties (deadlock-freeness and liveness) are preserved by the decomposition. In that case, we can reduce the problem of checking that a system holds these properties to the study of the properties for each Petri net (main and spawned goals).

There are several studies that deal with some kind of hierarchical Petri nets [39] and Petri net refinements [40,41]. In all these cases, each refinement step in the hierarchical representation consists on the replacement of one transition by a subnet that represents its behavior in more detail. In the solution proposed here, the spawn goal structure defined in Section 3 is replaced by the goal PN. The spawn goal structure in this case is represented by a place (execution of the subgoal) and a transition (end of subgoal execution) instead of only a transition.

### 5.2. Terminability Analysis

Petri nets have been used to model and analyze different kinds of systems such as communication protocols and factory processes where liveness is a very important property. However, in robotic task-modeling systems, most tasks and subtask should finish. For that kind of systems, the terminability property plays a very important role.

**Definition** **1.**
*A terminable Petri net (TPN) is a Petri net with a final marking. This is, TPN=(P, T, F, $M_{0}$,$P_{f}$) where:*

*$P,T,F,W,M_{0}$ were defined in Section 3.2.*

*$P_{f}$ are the final places, with $|P_{f}|\geq 0$ (i.e., {$\emptyset \subseteq P_{f}$}). Whenever all tokens of current marking $M_{i}$ are within the final places $P_{f}$ ($|M_{i}\cap \left(P-P_{f}\right)|=0$) or there are no tokens left at all ($|M_{i}|=0$) the execution of the TPN terminates.*



These final places are a new feature introduced in the model and allows the system to control the actual termination of any task. Graphically, all the final places in $P_{f}$ are represented by a double border circle. For example, in Figure 8 place labelled as “End B” is the only place for Task B in the final places $P_{f}$ ={$P_{EndB}$}. Therefore, whenever there are no tokens in the net or there is only one token in place “End B”, the execution terminates.

**Definition** **2.**
*A marking $M_{t}$ is considered a terminal marking if fulfills the termination condition: {$M_{t}\subseteq P_{f}$}. This is, all tokens of current marking $M_{i}$ are within the final places or there are no tokens left at all, given that {$\emptyset \subseteq P_{f}$}. Whenever the execution of a net reaches a terminal marking, it will terminate.*


**Definition** **3.**
*A Hierarchic Terminable Petri net HTPN is a terminable net that might include terminable subnets. The hierarchical tree of a net A represented as $H_{A}$ is the tree of nets under A. This is, all the PNs that are started directly or indirectly from A including A.*


The execution of a PN (subtask B) inside another one (task A) is represented as a sequence of a place and a transition (Figure 8). When the first transition is fired ($t_{2}$ in Figure 8), the subnet (Task B) will start with the initial marking. The place in the main task (labeled “Task B” in Figure 8) would have a token while the subnet is running. The condition associated to the final transition ($t_{4}$ transition in Figure 8) is that the subnet (Task B) reaches a terminal marking $M_{t}$.

**Definition** **4.**
*Terminability condition $\tau^{s}$: A TPN holds the terminability condition if for any marking ($M_{i}$) derived from the initial marking {$M_{i}\in R\left(M_{0}\right)$} exists some sequence of firings $R(M_{i})$ that ends in a terminal marking of the TPN {$m_{t}=R\left(M_{i}\right)$}*
(1)$$\tau_{s}\Rightarrow \forall M_{i}\in R\left(M_{0}\right),\exists M_{t}\in R\left(M_{i}\right)/M_{t}\subseteq M_{f}$$


This condition grants the termination of a net regardless of the marking evolution, i.e., it is free of deadlocks and some termination marking is always reachable.

Since we work with interpretable Petri nets, it is very important to notice that all transitions of the net must be firable to guarantee the terminability condition of the net. This is, it might exist a firing sequence that ends in a terminal marking. However, if there is a transition in the firing sequence that cannot be fired, the TPN will never terminate.

**Theorem** **1.**
*If all the nets in a hierarchy tree $H_{A}$ hold the terminability condition, then the task A is also terminable.*


**Proof.** Since the main TPN associated to task A is terminable, there is at least a sequence that leads to a terminal marking. If that sequence includes some transition associated to a subnet, we have to make sure that the transition is firable. In other words, the condition associated to the transition will eventually become true. According to Definition 3, the condition associated to the subnet transition is the termination of the subnet. However, since all the subnets hold the terminability condition, can eventually terminate and the condition will eventually become true. □

From the last theorem we can determine that the task represented by the hierarchic net A is free of deadlocks since it will always end regardless of the transition firing sequence of the main net and subnets.

According to Theorem 1, we need to analyze all the PNs in the hierarchy ($H_{A}$) in order to find out if they hold the terminability condition. For this analysis, we propose the use of the state space represented by the Coverability Tree [30].

In the case of bounded nets, the Coverability Tree is called Reachability Tree (RT) [30] since it contains all possible reachable markings. The coverability tree is a tree representation of its possible firing sequences. Nodes represent markings generated from the initial marking $M_{0}$ (root) and its successors as shown in Figure 9. Each arc in the tree represents a transition firing, which transforms one marking to another. The Reachability Graph (RG) is obtained from the RT by merging the nodes with the same marking as shown in Figure 9. The RT can be obtained by the Karp and Miller algorithm [42] that for the case of binary Petri nets is reduced to Algorithm 1.

**Algorithm 1** Algorithm to obtain the reachability tree1: Label initial marking $M_{0}$ as the root of the tree and tag it as *new*.2: **for** each *new* marking **do**3:  select a new marking M.4:  **if** M is identical to a marking on the tree **then**5:   tag M as *old*6:   go to another *new* marking7:  **end if**8:  **if** no transitions are enabled at M **then**9:   tag M as *dead-end*.10:   **end if**11:   **for** each enabled transitions at M **do**12:    obtain the marking M’ that results from firing t at M.13:    introduce M’ as a node, draw an arc with label t from M to M’ and tag M’ as *new*.14:   **end for**15: **end for**

**Theorem** **2.**
*In order to verify the terminability of a single TPN, its RG must fulfill the following:*

*All markings at the end of branches must be terminal markings (covered by the final marking $M_{f}$).*



**Proof.** Since the RG represents all possible firing sequences, the terminal nodes represent all the possible markings where it can stop. If the markings of all those terminal nodes are terminal markings it fulfils the terminability condition. □

In the example of Figure 9 the Reachability Tree has three leave nodes. The leave on the left has marking $M_{7}$, which is a terminal marking according to Definition 2. The other two leaves have markings ($M_{4}$ and $M_{6}$) that are included in the left branch and therefore hold Theorem 2.

Even though this is an exhaustive method, the hierarchic decomposition allows for small nets whose reachability tree can be easily obtained with the aforementioned algorithm. In Section 6 we will quantify the time and memory reduction obtained with the hierarchic decomposition.

### 5.3. Liveness Analysis

Some of the tasks that an autonomous robot system can execute can also be cyclic. For example, a surveillance robot can execute a cyclic patrol task. The analysis of that kind of Petri nets is very similar to the nets on other applications such as discrete process control in manufacturing systems. For that kind of systems the Petri nets are desirable to be deadlock-free and live. If a net is live it is known to be free of deadlocks, as every transition can always be fired again, no matter the firing sequence [30].

**Theorem** **3.**
*If all the nets in a hierarchy tree $H_{A}$ hold the terminability condition and the main Petri net holds the liveness condition, then the task A is live.*


**Proof.** This theorem is a direct consequence of applying Theorems 1 and 2. □

From the last theorem, we can determine that the task represented by the hierarchic net A is live and therefore also free of deadlocks. This means that every transition within A will be always fireable again and its subnets, if any, will not block its evolution. Also, every transition within every subnet (at every level) are also always fireable again. In this case, the hierarchic system of TPNs $H_{A}$ works similar as a bigger single live net.

As in the terminability case, the liveness analysis of the main PN is based on the state space constructing the RG. First, the PN of the subtasks are analyzed for terminability as described above and second, the liveness of the main (root) PN needs to be analyzed.

A possible method to analyze liveness from the RG can be derived from former publications such as [43]. The nodes (possible markings) of the RG can be partitioned into strongly connected components (i.e., maximal set of mutually reachable states). For liveness, the *ergodic* or *absorbing* components are the maximal set of states (nodes) mutually reachable so that there are no paths going out of the component. Once constructed the ergodic components, a transition *T* is live if and only if an arc *t* appears in all the ergodic components. The PN is live if all the transitions are live.

The Petri net for Task A (Figure 9) is not live because is terminable and liveness and terminable are mutually exclusive. However, a live version of Task A (Task A’ in Figure 10) can be obtained changing the termination condition of place $P_{6}$ and adding a loop back to the initial place ($P_{0}$). For the RG of Figure 9, the *ergodic* component is the hole RG since it is not possible to make a smaller set of nodes mutually reachable. Since the RG includes all the transitions, all of them are inside the unique *ergodic* component and therefore all of them are live.

## 6. Results

TDL is processed using a compiler that transforms the task definitions into pure C++ code that includes calls to the TCM library. As a result of this research, a task analyzer has been added to this parser. The task analyzer provides two main functions to the system:A *parser* from TDL to PN that translates every Goal, Command, Monitor and Exception handlers into a Petri net and stores them in a PNML format.An *analyzer* that analyzes each Petri net and reports the results in a text format.

The main advantage of using the PNML format as the output of the parser is that there are several Petri net tools such as PIPE [32] that are able to read these files and represent them in a graphical interface. Some of those tools include also a wide variety of modules for different kind of analysis. However, due to the nature of the tasks modeled here and the hierarchy decomposition proposed, new properties and the corresponding analysis have also been implemented. This research includes methods to set the terminability and liveness properties of a task from the individual analysis of the Petri nets in the hierarchy tree of the task.

### 6.1. Problems Detected by the System

This system has been tested with several TDL programs. One example of the kind of problems that we detect is the deadlock in the *centerOnDoor* goal for the TDL program shown in Listing 4. A deadlock-free definition of this goal was already presented in Listing 1.
Listing 4: TDL Goal *centerOnDoor* with a deadlock.
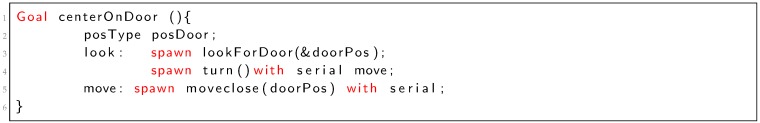


In Listing 5 we can see the output of the analyzer that reports a deadlock and shows the shortest way to it in two different formats. The first one is the sequence of transitions that are fired before reaching the deadlock. This format is the easiest way to find the deadlock looking at the Petri net. In the example, from the PN of Figure 11 we can see that from the initial marking, after firing transitions $T_{6}$, $T_{0}$ and $T_{1}$, the system gets stuck. The second format shows the sequence of tasks that are executed before reaching the deadlock. This format is intended to guide the programmer with the TDL code. In the *CenterOnDoor* example (Listing 4), goal *lookForDoor* is the only one that can be executed before the deadlock.

It is important to notice though that when a TDL task-control program is running in default mode, it can detect the deadlock and try to avoid it. The resulting execution trace however might be quite different from the one expected.
Listing 5: Analyzer output for the *centerOnDoor* goal of Listing 4.
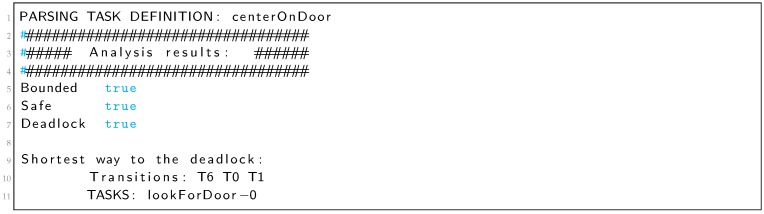


The Petri net obtained for the *centerOnDoor* goal in Figure 11 shows how the constraints between the *turn* and *moveClose* goals deadlock the system.

Another kind of situation that can be detected with this system is illustrated by rewriting the *goTo* goal as in the Listing 6. Listing 7 shows the report of the analyzer that includes the deadlock and the shortest way to reach it. The Petri net obtained in this case (Figure 12) shows that the constraint on task *navigateTo* is on a goal executed in one branch of an **if**. That goal can be pending forever if that branch of the **if** is not executed.
Listing 6: Goal *goTo* with a possible deadlock.
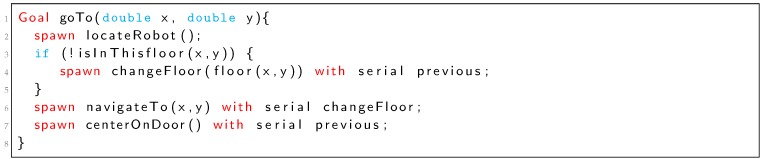

Listing 7: Analyzer output for the *goTo* goal in Listing 6.
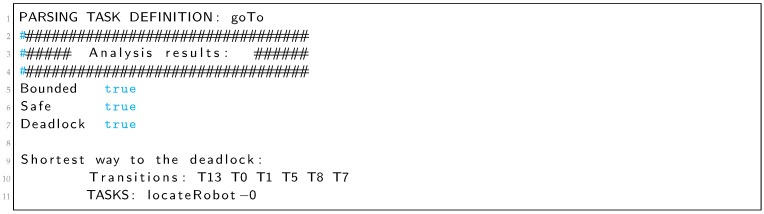


As in the first situation (deadlock in Listing 4), it is also important to point out that TDL would detect this situation and would remove the constraint in order to avoid the deadlock.

### 6.2. BellBot Application

The analyzer resulting from this research was applied to the BellBot application developed in the UVigo robotics lab [12]. This is an automatic hotel assistant system based on a series of mobile platforms that interact with guests and service personnel to help them in different tasks. The robot shown in Figure 13 was built on a four-wheel synchro-drive base and has independent control over translational and rotational velocities. Several sensors including tactile, sonar and laser sensors enable interaction with the physical world.

In the first version of BellBot, tasks were programed using RoboGraph [33]. For the research presented here, we implemented a new version in TDL to test the parser and the analyzer. The tasks include bringing small items to customers, showing them different points of interest in the hotel, accompanying the guests to their rooms and providing them with general information. Each robot can also autonomously handle some daily scheduled tasks. Apart from user-initiated and scheduled tasks, the robots can also perform tasks based on events triggered by the building’s automation system (BAS). The robots and the BAS are connected to a central server via a local area network. The system has been tested in the ETSII building of the University of Vigo. Faculty offices were used as guests’ rooms, one lab was used as the bar and the lobby was used as the reception area. The BellBot web page [44] shows some videos with the robot executing some of the tasks and some user interactions.

Verifying the tasks was included as another step in the programming process. Most of the problems detected were due to the task hierarchy since sometimes the programmer does not look at the situation as a whole when defining relations between tasks on different levels.

### 6.3. Efficiency Analysis

The main problem with the analysis method proposed is that the number of nodes in the Reachability Tree can grow exponentially with the size of the PN. The growth rate is greater for PNs that have many parallel running threads. However, with hierarchy, the analysis takes place in a set of small PNs instead of analyzing a big PN. A possible way to illustrate the benefit of using the hierarchy in the analysis step is to evaluate the difference between analyzing the system with and without hierarchy.

To get an idea of the relation between the size (number of places and transitions) of the PN and resources (time and memory) needed for the analysis as proposed in Section 5, we have conducted the analysis on a series of PNs with different sizes. We started with a simple PN, conduct the analysis and register the time and number of nodes in the RT (memory). Then, we add more places and transitions, conduct again the analysis and register again the results. Table 1 and Table 2 show the increase of the RT nodes and analysis time as we keep adding places and transitions to two different PNs. Both tables show a different increase rate mainly because of the different parallel threads (branches) in the PNs. The PNs on the sequence for the second case (Table 2) include more parallel threads than the first one (Table 1) and therefore there is a greater increase on the number of possible combinations of markings producing a superior increase in the number of RG nodes and consequently the time analysis.

Nevertheless, in both cases the CPU time and memory increases exponentially with the complexity of the PN (number of places and transition). Therefore, dividing a PN in small PNs will simplify the analysis from the point of view of time and memory.

To get an idea of this benefit for the BellBot case, we analyze the case of one of the main Tasks that include 11 PNs. First, we performed the analysis with the 11 PNs. Then, at each step, we merge some PNs by inserting them in the places where they were used and repeat the analysis. The results in Figure 14 show the big reduction of analysis time as the number of the PNs in the hierarchy increases. A simple change from one to three PNs entails a big reduction on the analysis time. The division has been carried out balancing the number of places and transitions so that all the PNs have a similar number.

## 7. Conclusions

This paper proposes a generic approach to the autonomous systems verification problem by focusing on high-level task definition languages. Many efforts have been made for verification of autonomous systems but they have little impact on the way these robotic systems are developed and validated. This is mainly because many solutions require the use of specific formalisms to define again and verify these systems. We have addressed this problem by developing tools that allow the direct analysis of the system behavior through the programs that the developer use to define the high-level tasks. The idea is to make the verification step part of the IDE in order to develop the autonomous systems. This involves making this step easy enough to use so that it can be part of the implementation process without extra work for the developers.

The process presented here includes two steps: a first step that automatically obtains the PN representation of the tasks and a second step that verifies the system by analyzing the PNs.

For the case implemented here, the high-level tasks are defined in TDL and a parser obtains the PN structures as described in previous sections. Results show that a wide range of possible problems can be detected without the tedious process of manually translating software into formal models. The report produced by the analyzer includes some counterexamples that show the programmer the possible sequence of events before reaching the problem.

The analysis of the PNs has been adapted to formally verify the tasks of autonomous robot systems. For that purpose, new properties have been added to the classic PN analysis. A method to carry out this analysis has also been defined and applied to the BellBot application.

As a summary, the main advantage of the solution presented here with respect to traditional validation systems is to use directly the task definition program. For that purpose, an automatic parser obtains the formalisms that model the system. Those formalisms are then used by the analyzer to establish some general properties such as liveness and terminability. Unlike model-checking systems, the solution presented here does not allow to define formal specifications that indicate desirable properties of the system.

## Figures and Tables

**Figure 1 sensors-19-04965-f001:**
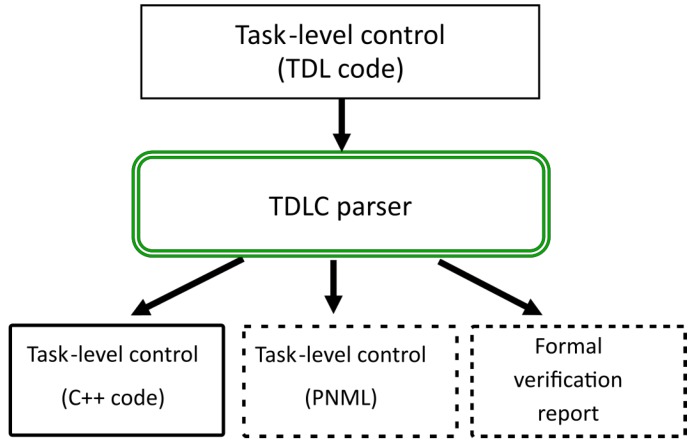
The Task Description Language (TDL) parser translates TDL code into: C++ code, Petri nets (PNML) and a formal verification report.

**Figure 2 sensors-19-04965-f002:**
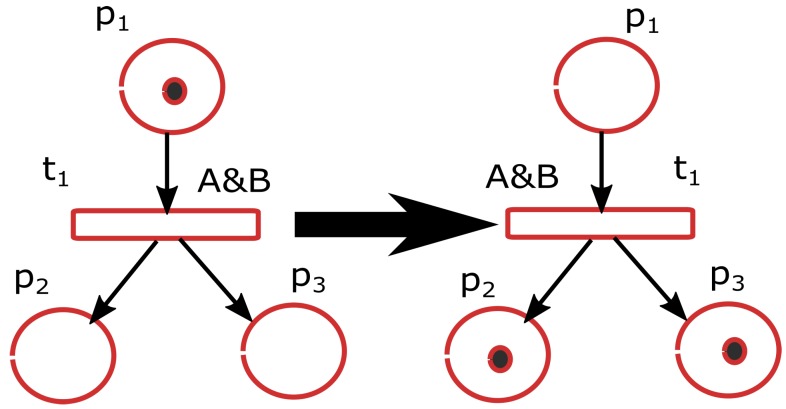
Marking evolution in a simple petri net with three places (circles) and one transition (rectangle). On the left, the transition is enabled because all the input places have a token. The marking of the petri net (PN) shown on the right is the result of firing the transition t1 of the PN on the left.

**Figure 3 sensors-19-04965-f003:**
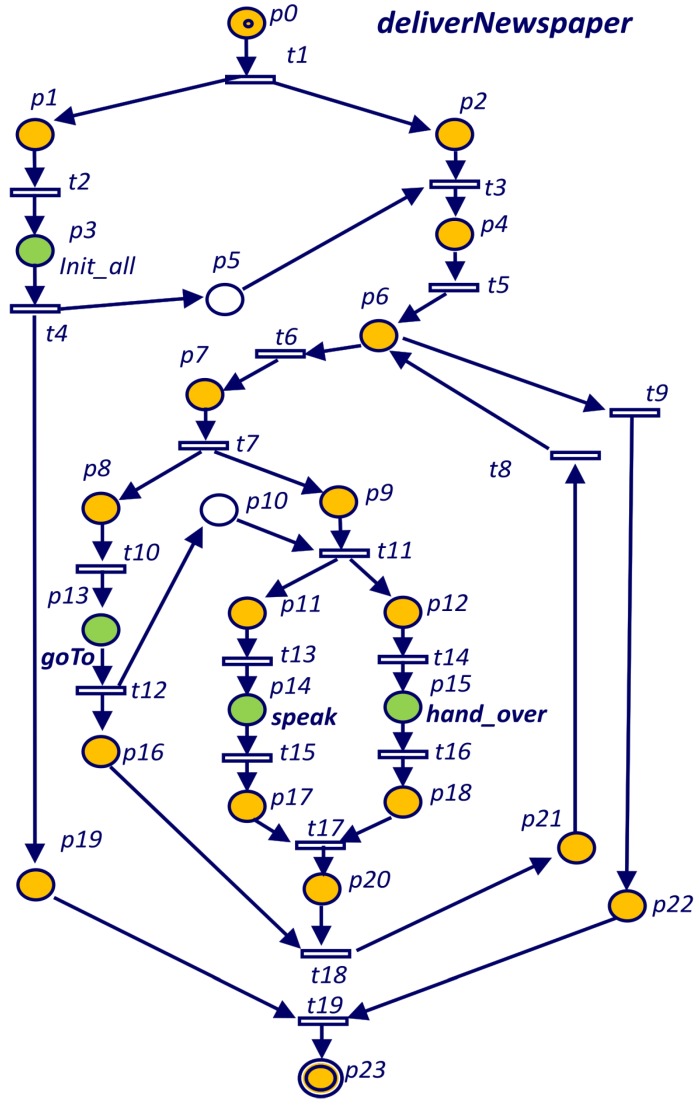
Petri net for the *deliverNewspaper* goal defined in Listing 1.

**Figure 4 sensors-19-04965-f004:**
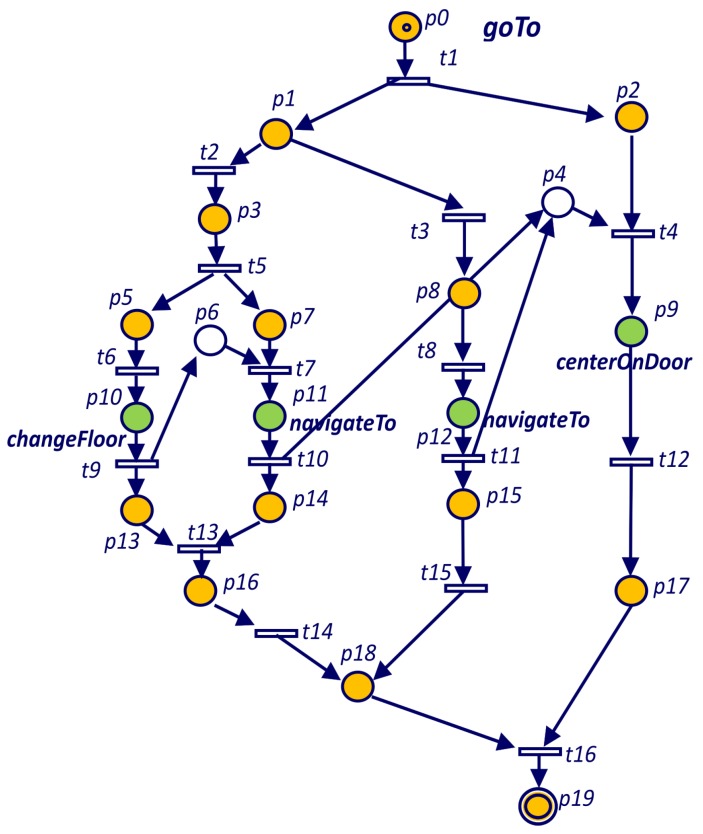
Petri net for the *goTo* goal defined in Listing 1.

**Figure 5 sensors-19-04965-f005:**
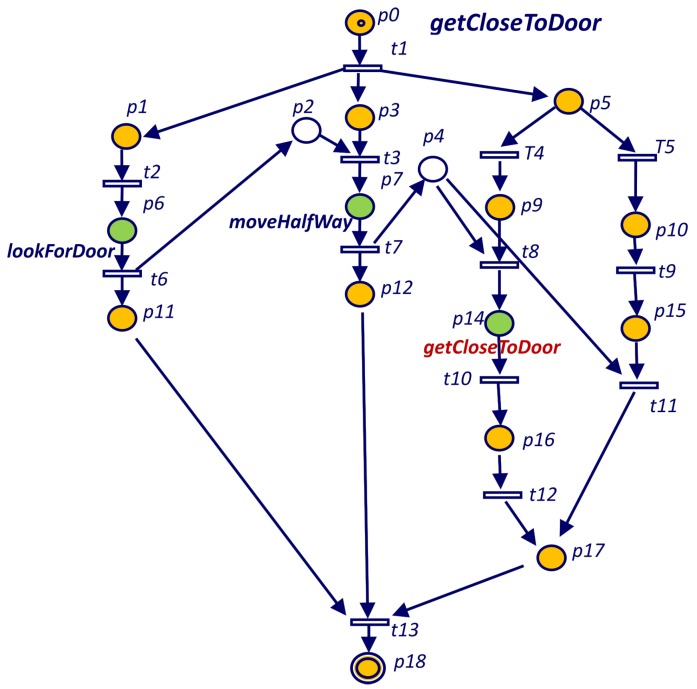
Petri net for the *getCloseToDoor* goal defined in Listing 2.

**Figure 6 sensors-19-04965-f006:**
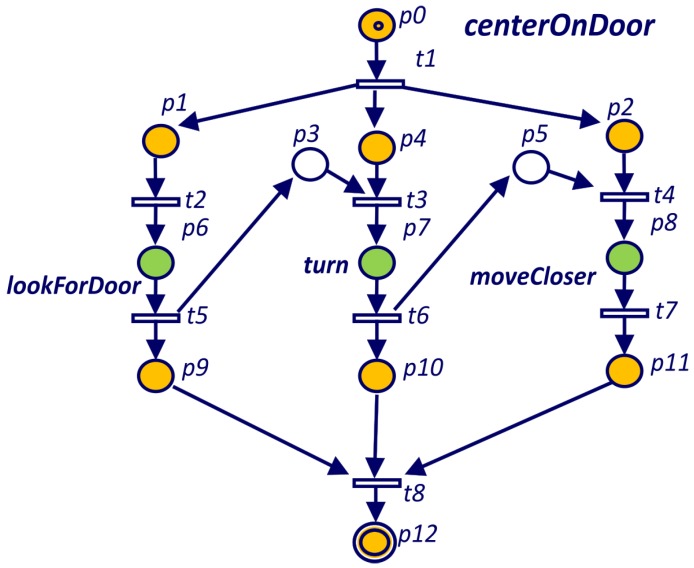
Petri net for the *centerOnDoor* goal defined in Listing 1.

**Figure 7 sensors-19-04965-f007:**
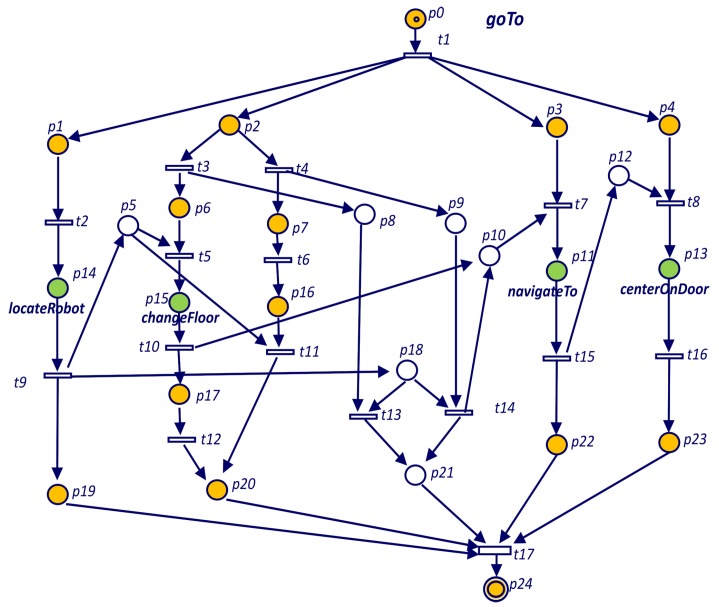
Petri net for the *goTo* goal defined in Listing 3.

**Figure 8 sensors-19-04965-f008:**
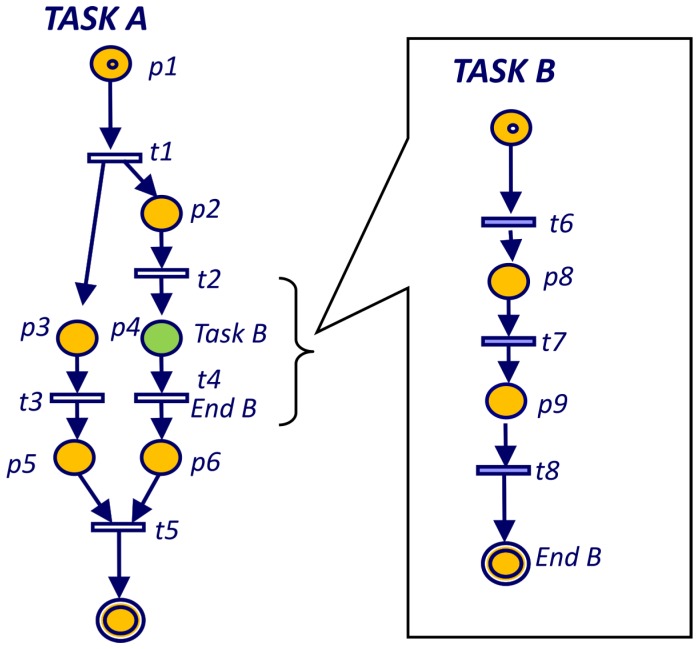
Hierarchical Petri net.

**Figure 9 sensors-19-04965-f009:**
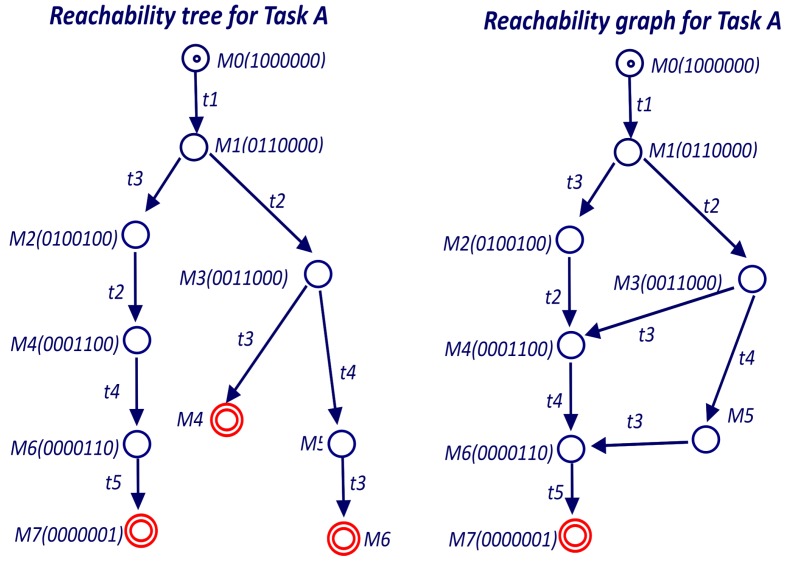
Reachability tree and graph for main PN of Task A.

**Figure 10 sensors-19-04965-f010:**
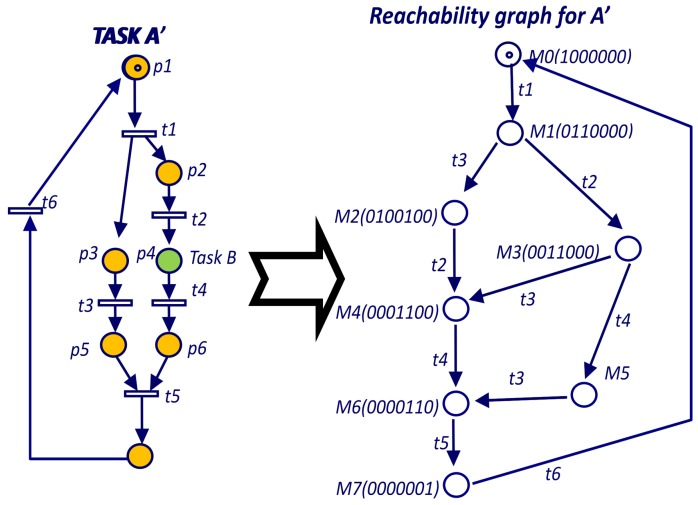
Reachability graph for main PN of Task A’.

**Figure 11 sensors-19-04965-f011:**
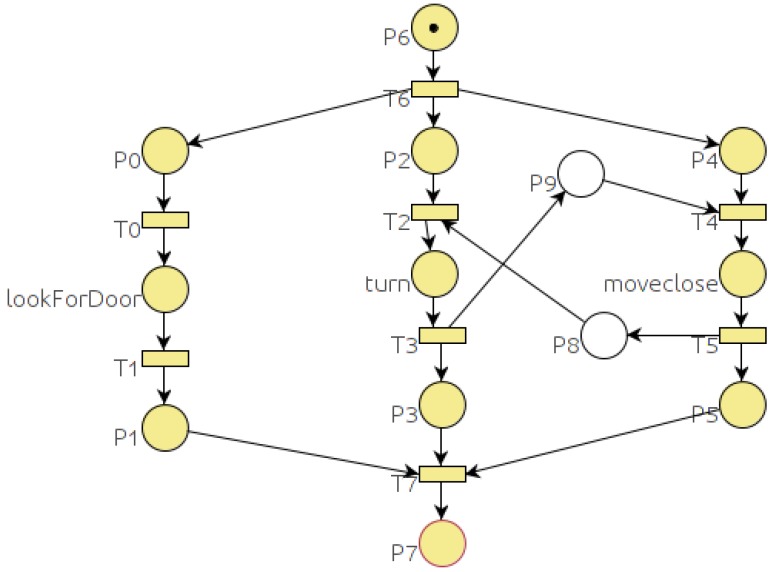
Output PN corresponding to the TDL goal defined in Listing 4.

**Figure 12 sensors-19-04965-f012:**
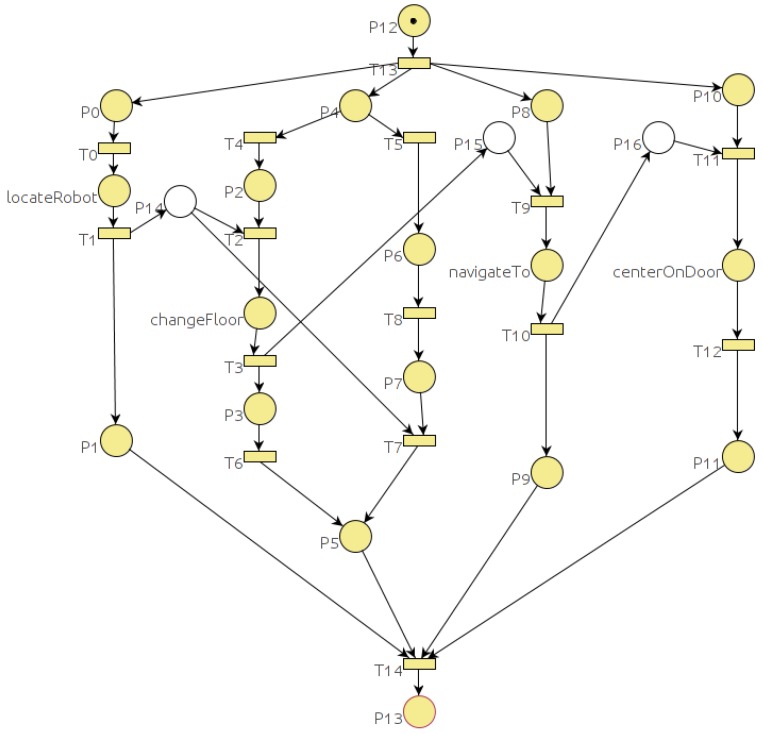
Output PN corresponding to the TDL goal defined in Listing 6.

**Figure 13 sensors-19-04965-f013:**
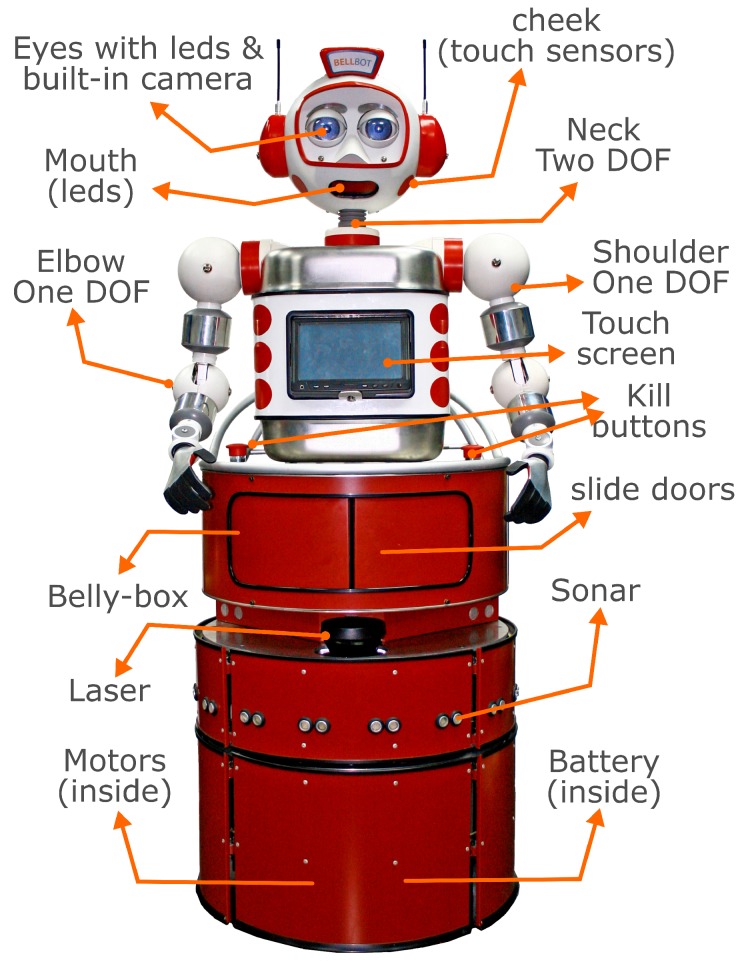
Hotel Assistant robot BellBot. The base includes two brushless Longway motors (80 W/24 V, 6.2 N/m), One LiFePO4 26 V/40 Ah battery that provides power autonomy for 3 to 5, bumpers (all the skin works as a bumper) and a Sick LMS100/10000 laser.

**Figure 14 sensors-19-04965-f014:**
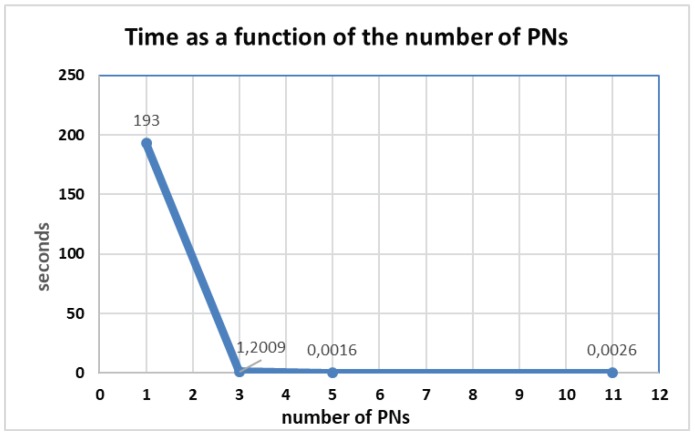
A main task with 11 subtasks was analyzed. First using the 11 PNs and then merging the PNs in five PNs, three PNs and finally only one PN. The analysis time has been registered for each of the four different cases and represented in the above plot.

**Table 1 sensors-19-04965-t001:** Every row registers the size of the PN (number of places and transitions) and the resources to conduct the analysis (time and RG nodes). In this case, the sequence of PNs have few parallel threads.

Places	Transitions	Nodes	Time (msec)
9	8	17	0.9
12	10	59	0.6
15	12	221	4.1
18	14	815	24.2
21	16	2921	166.3
23	18	10,211	1916.8
27	20	34,997	28,156.3
30	22	118,103	407,390.5

**Table 2 sensors-19-04965-t002:** Every row registers the size of the PN (number of places and transitions) and the resources to conduct the analysis (time and RG nodes). In this case, the sequence of PNs have more parallel threads than Table 1.

Places	Transitions	Nodes	Time (msec)
15	14	65	1
21	19	545	18.9
27	24	4325	457.4
33	29	32,405	34,884.75
39	34	233,285	2,607,840

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
