# Peer review of "Formal Verification for Task Description Languages. A Petri Net Approach"

_sensors, 2019, doi:10.3390/s19224965_

Round 1
Reviewer 1 Report
This is an interesting study apply formal verification methods for Autonomous robot systems which
require some language to support task-level control. Having said that, I think the authors should invest more time to strengthen this study.
Although this paper is written well, there are numerous grammatical errors which need to be fixed.
Below are just some of these mistakes:
Line 39, applications developed should be developed application
Lines 50, “The first step in this process consists is” is a wrong grammar
“he TDL language was selected for because” wrong structure
Line 75, “challenge” should be “challenging”
The contribution of this study should be highlighted in the abstract section, I know you talk about it briefly but real contributions needs to be highlighted
A comparison and contrast should be made between a method such as reinforced learning and this method. What makes you think the benefit of your method over other unsupervised learning such as reinforced learning, or how you method could be implemented for those methods.
This study consists of more than 20 pages so having a solid structure helps readers to follow up on the content of this manuscript, A comprehensive flowchart talking about the flow of this paper is needed. What is the difference between “Related work” and “Background”?more works need to be spend and maybe you could shorten this paper!
Figure 1 description is confusing, might be unnecessary to describe everything in the description
The contribution of this study should be highlighted at the end of the background section, what do you have to offer
Many short lines throughout the manuscript, look at page 15, for instance.
A weak conclusion section.
Reviewer 2 Report
The author presents an effective approach to apply formal verification methods for the language to support task-level control. The approach translates the task-level control models into a Petri-Net (PN) based representation. It is used to define new methods to analyze some properties of the Tasks such as liveness, deadlock-free and terminability. They also applied their approach to the Task Description Language (TDL) to illustrate its performance by experiments.
I suggest that the submission may be accepted, but only after some major revisions have been made to the manuscript. My main concern is how to prove the advancement of this work. It is recommended to add a comparison experiment with the current mainstream methods to show that the method is better or slightly better than the traditional method.
In addition, the author should provide a clearer, more informative discussion to help the reader understand 1) what specifically are the advantages of this proposed methods relative to previous algorithms? Are these differences substantial (i.e., is this method substantially better) or relatively minor? 2) what are the limitations of this particular method? Are they similar to limitations of the previous approaches? What future research steps would be most beneficial to address these limitations?
Reviewer 3 Report
There are several suggestions to improve this paper.
The definition or references for “liveness, boundedness, deadlock free” is needed. “For example, [23] uses PNs to model single-robot tasks”, this reference style is not preferred. The first name of the authors of [23] should be mentioned. Some parts of this paper used the first person in narration, which is not normally used the words like "we…".
Reviewer 4 Report
The paper describes verification of robotic systems using TDL converted to Petri Nets. In Petri Nets, model checking technique is applied. Petri Nets are defined formally, but formal definition of TDL and the rules of conversion are not addressed in the paper, only brief description is given with simple examples. Formal definitions of TDL and conversion should be provided.
The authors discuss some time-dependent (including absolute time) issues of verified systems, and they refer to Uppaal and Prism, but this issue is not covered in the paper. These issues should be addressed.
The paper claims specification and verification od asynchronous systems, but asynchrony is not discussed as a feature of verified systems and of presented model. Also, there are no references to other papers concerning verification of asynchronous systems, for example papers on verification of Karlsruhe Production Cell benchmark:
Björnander, S., Seceleanu, C., Lundqvist, K., Pettersson, P., 2011. ABV - A Verifier for the Architecture Analysis and Design Language (AADL), in: 6th IEEE International Conference on Engineering of Complex Computer Systems, Las Vegas, NV, 27-29 April 2011. IEEE, pp. 355–360. https://doi.org/10.1109/ICECCS.2011.43
Daszczuk, W.B., 2019. Asynchronous Specification of Production Cell Benchmark in Integrated Model of Distributed Systems, in: Bembenik, R., Skonieczny, L., Protaziuk, G., Kryszkiewicz, M., Rybinski, H. (Eds.), 23rd International Symposium on Methodologies for Intelligent Systems, ISMIS 2017, Warsaw, Poland, 26-29 June 2017, Studies in Big Data, Vol. 40. Springer International Publishing, Cham, Switzerland, pp. 115–129. https://doi.org/10.1007/978-3-319-77604-0_9
Minor comments:
l.48. deadlock free -> deadlock freeness
l.50. process consists is -> process is
l.52. selected for because -> selected because
l.80. many different architectures – of what ?
l.81. This research - what research ?
l.82. full stop missing
l.95. UPPAAL – previously Uppaal
l.117. this process – what process ?
l.131. TCM – shortcut not expanded
l.146. Once… - the sentence without a subject
l.146. init_all - boldface is reserved for keywords. Italics should be used or the font identical to the listing. This comment applies to consecutive boldface identifiers.
l.148. The last sequence of three tasks are… - where are the tree tasks in the example ?
l.171. tokens – not defined
l.174. marking – not defined
l.176. The input places of a transition t … is defined -> “The set of input places ... is” or “The input places ... are”
l.179. On -> In
l.179. the output places of a transition t … is -> “the set of output places ... is” or “the output places ... are”
l.205. The execution of a task refers to the time it takes to handle all leaf-node tasks. - in a subtree ?
l.211-212. Actions can include conditional, iterative, and even recursive code. - Please give examples, referring to the example above
l.213-214. Each task tree represents a single execution trace of the control program. - Please give an example in listing 1
l.214. from the formal verification point -> from the formal verification point of view
l.220. analyzed independently - why independently ? Aren’t there any dependencies between them ? If so, why aren't they separate programs ?
l.236. the goal ends - ends = is reached ?
l.236. a new PN is created - PN has not "create" operation in its definition. What are the relations between existing PN and a created one ?
l.243. that requested the newspaper - not request is modeled in the listing. it seems like active searching of the target by the robot (getNextRoomCoordinates).
l.246. If the while condition evaluates to true - the transition is unconditional. The conditions should be introduced externally to PN. Please describe how.
l.247. the tasks goTo, - goTo task is not named in Fig. 3
l.259. an empty place - What is an "empty place" ? What is an "action" in PN ?
l.271. executed in parallel - in PN, typically interleaving semantics is assumed. This should be commented.
l.271-272. parallel constraints - what are parallel constraints and why they need not be modeled ?
l.274-275. transition t3 is not enabled until place p3 has a token, which means that turn goal has finished - is it general, that two spawn statements in sequence require a place like p3?
l.278-279. task centerOnDoor of listing 1 labels “look” and “move” serve this purpose - there are no multiple tasks of the same name
l.289-290. Place p5 represents the serial restriction between locateRobot and changeFloor (if condition evaluates to true) or between locateRobot and navigateTo - I see that p5 is connected with unnamed branch right to changeFloor, rather than with navigateTo
Figure 6. The third branch from left (t4,p7...) is not named
l.313. modelling - previously "modeled" was spelled with single 'l'
l.314. systems can -> systems that can
l.316. modelled - previously "modeled" was spelled with single 'l'
l.320-321. This can be seen as hierarchical PNs where the hierarchy is used to break down the complexity of the model - a reference on hierarchical PN is needed.
l.321. modelled - previously "modeled" was spelled with single 'l'
l.327. there exists a sequence of transition firings that grants ti can be enabled and fired again - I think that this conditions holds for cycling PN. In terminating PN a transition can be executed once or finite number of times, without losing its liveness.
l.332-333. However, liveness is a more restrictive condition than deadlock-free.
l.333. deadlock-free -> deadlock-freeness - Please explain why
l.354. |Pf| - is this cardinality?
l.354. curly braces should not be in italics
l.361. the execution terminates - deadlock are defined locally (i.e. in terms of individual transitions), while termination is defined globally: all tasks that are supposed to terminate, must terminate (in a subnet or the global net).
l.367. A including A - from this point the text should not belong to the definition
l.376. guaranty -> guarantee
l.401. mf -> Mf
l.433. an arc t appears -> ‘t’ should be in italics
l.442-443. code that include -> code that includes
l.445. Goal, Command, Monitor and Exception handlers - the analysis of the latter three is not covered in the paper
l.476. goal can be pending forever if that branch of the if is not executed - the sequence T13 T0 T1 T5 T8 T7 does not affect centerOnDoor task. Please explain
l.477. As in the first situation - which situation is first ?
l.478. removes -> would remove
references – DOI identifiers are not provided
English should be improved
Round 2
Reviewer 1 Report
The authors addressed most of my comments properly. Having said that, the edition portion has not been addressed and unfortunately i am going to reject the paper if they did not address well.
For instance, " iff " line 449.
or
"From the last theorem, we can determine that the task represented by the hierarchic net A is live 437 and therefore also free of deadlocks.
438 This means that every transition within "
Line 438 is coming after line 438 and it should not come in a new paragraph.
These are basic things that i should not be reiterated.(this should be taken care of through the whole manuscript)
You said " The contribution you are talking about was included at the end of the “Introduction” section" where is that?
Again i just mentioned some editing needs to be done and without extensive revision this paper cannot be accepted for publication.
Reviewer 2 Report
All my previous comments have been incorporated in the revised version. But the text should be revised and edited by someone with professional English editing experience and also careful proofreading is needed.
Reviewer 4 Report
The most important issue in the first review remains unaddressed: the lack of formal definition of both TDL and translation rules from TDL do PN. Only brief descriptions are given with simple examples.
l.221-222. Actions can include conditional, iterative, and even recursive code. – Examples of the former two are given, referring to the example, however the recursive one is most interesting, yet no example is given.
l.348 deadlock-free -> deadlock-freeness
